# Sequential islet transplants for type 1 diabetes are not associated with sustained or cumulative increases in hepatic portal venous pressure

Alice L. J. Carr[1]*, Rahi Shah[2], Braulio A. Marfil-Garza[3], Anna Lam[1,4], Khaled Dajani[4], Blaire Anderson[4], Richard T. J. Owen[5], Doug O'Gorman[4], Tatsuya Kin[4], David Bigam[4], A. M. James Shapiro[4], Peter A. Senior[1,4]

1 Alberta Diabetes Institute, University of Alberta, Edmonton, Canada, 2 Royal College of Surgeons in Ireland, University of Medicine and Health Sciences, Dublin, Ireland, 3 Tecnologico de Monterrey, Institute for Obesity Research, Monterrey, Nuevo Leon, Mexico, 4 Clinical Islet Transplant Program, University of Alberta, Edmonton, Canada, 5 Radiology and Diagnostic Imaging, University of Alberta, Edmonton, Canada

* acarr1@ualberta.ca

## Abstract

Islet transplantation (ITx) is advancing rapidly, with clinical trials of stem cell derived islets demonstrating short-term insulin independence. However, long-term insulin independence may still require multiple infusions. We examined the effects of sequential ITx on portal venous pressure and factors influencing acute pressure changes across 693 intraportal ITx procedures in 298 adults (44% M); who received up to 5 procedures, at the University of Alberta Hospital over 24 years. We assessed acute portal pressure changes per infusion, using linear mixed-effects models to compare sequential pre-infusion pressures to baseline and assess relationships between pressure changes and packed cell volume (PCV), purity, islet dose (IE/kg) and estimated total liver volume (eTLV). Portal pressure transiently increased, similarly, after each infusion by an overall median 2.0 mmHg (IQR 1.0,4.0). PCV exhibited the strongest relationship with change in pressure, with 1mmHg increase per 1 mL of PCV, when adjusted for other parameters (Coefficient = 1.0 [95%CI 0.81,1.21]; p < 0.0001). Conversely, higher purity and larger eTLV were associated with smaller increases in pressure. A significant interaction term indicated that the effect that PCV has on pressure change may be moderated at higher purities. Our work provides insights from deceased-donor intraportal ITx that can inform the design of future clinical protocols for ITx.

## Introduction

Islet transplantation (ITx) is an established treatment for select individuals with type 1 diabetes in whom severe hypoglycemia impacts quality of life and poses significant threat to overall health and prognosis [1–3]. The procedure involves percutaneous

**Data availability statement:** Our Human Research Ethics Board requires that patient confidentiality is maintained. Due to the uniqueness of islet transplantation, the relatively low numbers receiving this treatment per year and the known location of the University of Alberta's Clinical Islet transplant program, there is a risk that the data underlying the results may contain potentially identifying information. Therefore, the data underlying the results reported in this Article are not publicly available. De-identified individual participant data, as well as a data dictionary, can be made available to researchers who provide a methodologically sound proposal with ethics approval. Proposals should be directed to the Clinical Islet Transplant Laboratory cil@albertahealth-services.ca with reference to this Article; to gain access, data requestors will need to sign a data access agreement.

**Funding:** The author(s) received no specific funding for this work.

**Competing interests:** BAMG is a co-inventor for a patent on TNFRSF25-mediated treatments of immune diseases and disorders (PCT/US2020/053085). AL has received consulting fees from Vertex. PAS has received grants from JDRF, CIHR and Brain Canada and consulting or personal fees from Abbott, Bayer, Dexcom, Eli Lilly, GSK, Insulet, Novo Nordisk, Sanofi, Vertex. He served previously as board chair for Diabetes Canada and is a co-lead for Diabetes Action Canada's innovations in type 1 diabetes goal group. AMJS has received grants or contracts from the Breakthrough T1D (formally Juvenile Diabetes Research Foundation), Canadian Stem Cell Network, Diabetes Research Foundation in Canada, ViaCyte, and the US National Institute of Diabetes and Digestive and Kidney Diseases; serves as a consultant to Vertex Inc, Betalin Inc, and Aspect Biosystems Inc; and is a co-inventor for a patent on TNFRSF25-mediated treatments of immune diseases and disorders (PCT/US2020/053085) and for a Cellular Transplant Site- Device-less technology (US 14/863541, CA.286512). All other authors declare no competing interests. This does not alter our adherence to PLOS ONE policies on sharing data and materials.

infusion of isolated and purified islets from deceased-donor pancreata into the portal vein of a recipient with type 1 diabetes. Unlike solid organ transplantation, most ITx recipients receive supplementary infusions from different donors throughout their follow-up [2].

While insulin independence can be achieved in almost 80% of recipients after infusion of islets from 2 or 3 donors, a gradual waning of graft function is common such that fewer than 10% of recipients remain insulin independent at 20 years [4]. A strategy of providing supplementary islet infusions has allowed individuals to regain and maintain long term insulin independence, with little additional risk associated with further procedures [5].

Although there is much interest in delivering islets to extrahepatic sites, the performance of intrahepatic ITx, delivered via the portal vein, in clinical settings is consistently superior [6] with an acceptable safety profile. The major procedural risks of this minimally invasive procedure are 1) intrahepatic or intraperitoneal bleeding from the hepatic puncture, and 2) thrombosis in the portal venous system. The University of Alberta Clinical Islet Transplant Program (CITP) have previously described strategies to mitigate these risks (anticoagulation and ablation the needle tract), which have been implemented by our center since 2005 [7,8]. In addition, the use of smaller volumes of purified islets in allogenic ITx have been associated with much lower risks than earlier experiences with autologous ITx, following total pancreatectomy, which were associated with severe portal hypertension and esophageal varices [9] and other serious complications [10].

Although long term outcomes of allogenic ITx demonstrate a favourable profile for patient safety and sustained metabolic control [4,11–13], both acutely elevated portal pressures and elevated packed cell volume (PCV) of the transplantation preparation have been suggested as potential risk factors for portal vein thrombosis [7]. Early work from the CITP observed that although rises in portal pressure were transient, there was an increased risk for acutely elevated portal pressure for those receiving up to three infusions [14]. In subsequent work we observed that larger increases in the portal pressure after transplant was an independent predictor of portal vein thrombosis, increasing the risk by 17% [7], with smaller liver volume and larger PCV noted to be associated with larger acute increases in portal pressure [7,14]. Since 2005, our centre has followed recommendations to maintain a PCV of the transplanted preparation below 5 mL, in addition to maintenance of therapeutic anticoagulation post-ITx and track ablation, which have also been implemented by many centers to reduce the risk of portal vein thrombosis [7]. As such, only thirteen (N = 12/1108 Islet Transplant Alone, N = 1/285 Islet After Kidney) portal vein thrombosis events have been reported in the most recent Collaborative Islet Transplant Registry (CITR) report [15], predominantly occurring before 2011, when these recommendations were collated and published [7].

While the risk of portal vein thrombosis is indeed low for initial ITx procedures, there remain important questions around the risks associated with supplementary islet infusions, which might be periodically repeated to maintain sustained insulin independence over decades. Understanding these risks is important not only for

optimal clinical practice and informed decision making in deceased donor islet transplantation, but for the future clinical delivery of stem cell-derived islet transplants as a therapy for type 1 diabetes.

The use of stem cell-derived islets to overcome the limited supply of donor pancreata is advancing rapidly. Recent clinical trials of stem cell-derived islet cells delivered intraportally via a transhepatic puncture (i.e., as with current clinical ITx procedures) have demonstrated insulin independence at 1 year, by restoring physiologic, glucose dependent insulin secretion, when combined with systemic immunosuppression [16]. If insulin independence wanes over time, as seen in ITx with deceased-donor islets [4,17], supplementary infusions of stem cell derived islets may be required periodically to maintain normoglycemia (e.g., every 5–10 years). Experience from clinical ITx is essential to inform the implementation and safety of future stem cell therapies, and crucial in discussions with regulators as these therapies progress to market.

To best inform these discussions, we examined the effects of supplementary islet infusions on the change in portal pressure in our large single centre cohort, which includes individuals who have received up to five intraportal islet infusions, while also delineating how the modifiable transplant-related parameters: PCV, purity and islet equivalents per recipient body weight (IE/kg), and the non-modifiable parameter: estimated total liver volume (eTLV), interact to influence acute portal pressure changes.

## Materials and methods

This is a retrospective study of patients receiving intraportal ITx at the University of Alberta Hospital between May 1999 to April 2023. This study was approved by our institutional health research ethics board (PRO000001120). Written informed consent for the use of health data for research purposes was obtained from all participants.

Electronic medical records data were collected to assess portal venous pressure before and after each intraportal infusion, as well as the parameters of each transplanted islet cell preparation: PCV, purity and IE/kg. Portal pressure measurements were first accessed for this research purpose on 20th April 2023 and islet isolation parameters were first access for this research purpose on 8th May 2024. All data analysis was performed on anonymized data. As some of the authors are clinicians who provided clinical care to subjects, they may have been able to identify participants during data collection. Subsequently after data collection these clinician investigators may be able to identify participants based on age and date of transplant. We included "standard of care" intraportal transplant procedures with complete portal pressure measurements (before and after infusion) and parameters of islet cell preparation. One participant was excluded from analysis as the procedure was a planned laparotomy using unpurified islets from a pediatric donor and not "standard of care".

Pancreata from deceased organ donors located in Canada (after neurological or cardiac death) were recovered by Give Life Alberta – Organ Donation North (previously Human Organ Procurement and Exchange program) within Alberta Health Services or the Provincial Organ Donation Organization after informed consent from the donor family has been obtained. Consent may be withdrawn at any point during the donation process. The informed consent and organ donation processes are subject to the Government of Alberta's Human Tissue Organ and Donation Act or local Provincial Equivalent which requires donor anonymity. Information from donor health records are subject to provincial law and policy and, as such public access to health information must be requested through a formal process. The full act can be found here: https://open.alberta.ca/publications/h14p5.

Individuals who wish to become organ donors can indicate this by signing (witnessed) their provincial health card, or joining the organ donation registry, but ultimately a family member or next of kin will be required to sign a consent form saying they have been informed about, and agree with, the donation process, even if the person that died has signed the back of their Alberta Personal Health Card or registered. Families of deceased individuals who have not indicated in advance their wish to donate an organ may be approached by medical staff to seek family consent. The informed consent process can be accessed: https://myhealth.alberta.ca/alberta/Pages/organ-and-tissue-donation-consent-to-donate.aspx.

Minors may have been a part of this donation cohort. All consent is performed under specific policy requirements that protect vulnerable populations and outline who may give informed consent. The basic principles include: Requires

capacity (or alternate decision maker), Must be Informed, Must be Specific, Must be Voluntary, Requires Understanding, Must be Documented. The full policy statement can be found here: https://extranet.ahsnet.ca/teams/policydocuments/1/clp-consent-to-treatment-prr-01-05-procedure.pdf "

## Parameters of the islet cell preparation

Islets were isolated from deceased donor pancreata as previously described [18]. Islets were quantified following culture by counting and expressed as islet equivalents (IE), the standard unit for reporting variations in the volume of islets correct to 150 μm diameter IE, and islet particle number (IPN). IE are adjusted by recipient body weight at admission to give IE/kg. Islet particle index (IPI), which is an indication for the size of islets relative to a standard 150 μm diameter IE, was calculated as IPN divided by number of IE. Viability assays were performed through the staining of the final islet product with SYTO-13 and Ethidium bromide, with assessment performed under fluorescent microscopy (DMLB, Leica Microsystems GmbH, Germany). Viability is represented as an average of the percentage of membrane-intact islets (SYTO-13+) to all islets (SYTO-13 + / Ethidium bromide+) across ten consecutive fields of view, with the minimum requirement for clinical utilization being 70%. Viability assays were not implemented prior to February 2002. Purity was determined as the percent of islets compared to the exocrine tissue present in the islet preparation, in addition to percentage of trapped (in exocrine tissue) islets. PCV is measured as the volume of the tissue pellet post-centrifugation immediately prior to transplant. Our standard approach to assessing PCV is to visually estimate the volume of cellular material in the bottom of 50 mL conical tube following centrifugation at 1100 rpm for 1 minute and rounded to the nearest 0.5 mL.

## Assessment of pre- and post-infusion portal venous pressure

The transplant procedure has been described previously [2]. Briefly, isolated islets were prepared as previously described [2] by suspension in xenoprotein-free media with no more than 5 cc of PCV per 100 mL of suspension medium mixed with 70 U/kg of heparin (procedures prior to 2005 used 35 U/kg of heparin) and transplanted intrahepatically via the portal vein accessed by a percutaneous transhepatic approach [19]. Portal pressure was recorded (mmHg) using a pressure transducer (Medex, Hillard, OH) prior to islet infusion, post-islet infusion and post-rinse infusion. The procedure was not initiated if an initial pressure > 20 mmHg was observed, or were paused and assessed if pressures reached > 20 mmHg during the infusion. Change in pressure, from before to after the infusion, was calculated as a delta change. Portal pressures, pre- and post-infusion, were compared pairwise. Pre-infusion pressures at each consecutive infusion were compared by mixed effects models to the pre-infusion pressure at the first infusion, while controlling for the effect of repeated measures and individual-level differences.

## Factors associated with portal venous pressure change

Mixed effects models with a random intercept, were used to assess the relationships between change in portal pressure, the specific parameters of the islet preparation: PCV, purity and IEs as well as eTLV, with the random intercept accommodating individual-level differences in the response variable, portal pressure change, across repeated measures. eTLV was calculated based on body surface area (BSA) by using the formula: $eTLV = −794.41 + 1{,}267.28 × BSA$ [20], with BSA calculated using Mosteller's formula [21]. These parameters were chosen as potential predictors of change in portal pressure based on clinical experience and previous literature [7,14]. An iterative modeling approach was utilized to identify the best combination of predictors for changes in portal pressure. A correlation matrix was used to identify interrelationships with selected predictors to inform possible interaction terms. Initial models included individual predictors and progressively more complex models incorporated additional predictors and interaction terms. An exploratory visualisation of the predicted changes in portal pressure was created using the final model to explore the interrelationships between purity and PCV at a fixed eTLV, set to the mean eTLV observed in CITP islet recipients (1524 cm³) to represent an average adult

islet recipient. The visualisation spans a grid of all possible values for purity (range 10%−100%) and PCV (range 1 mL - 9.5 mL) displaying modelled predictions of portal pressure across these ranges.

## Statistical methods

Analyses was performed using R statistical software version 4.3.0 (Foundation for Statistical Computing, Vienna, Austria). Pairwise comparisons were made by Wilcoxon signed rank test. Mixed effects models were computed using the *nlme* package [22] while predictions and prediction intervals were calculated using the *bernr* package [23]. Pearson correlation coefficients were generated using the *stats* package. Contrasts of the estimated marginal means between mixed effect models were calculated using the *emmeans* package [24], using the Kenward-Roger method for degrees for freedom and adjusted using the Dunnett method for multiple comparisons. The overall quality of each nested models was assessed using the *performance* package [25], computing conditional R-squared values to evaluate the proportion of variance explained by both fixed and random effects (with the random effects assigned as the patient-level groups), as well as marginal R-squared values to capture the variance explained solely by the fixed effects. Additionally, Root Mean Square Error (RMSE) was calculated to measure the average prediction error, while the Intraclass Correlation Coefficient (ICC) quantified the proportion of variance attributable the grouping structure (patient-level groups). To compare model fit, weighted Akaike Information Criterion (AIC) and weighted Bayesian Information Criterion (BIC) were used (i.e., normalised), indicating the relative likelihood of each model being the best fit. Accounting for each of these metrics, a calculated performance score allowed for identification of the most appropriate final model. Likelihood ratio testing (ANOVA) was also used to iteratively compare nested models (initial simple models vs. progressively more complex models) and determine if the additional complexity did indeed result in the model of best fit, i.e., the final model. To reduce multicollinearity, the variables were centered before fitting the final model. Data is summarised by median and interquartile range where appropriate. Significance level was tested at an alpha of 0.05.

## Results

### Cohort characteristics

We included 298 adults who received a total of 693 intraportal ITx procedures (Fig 1). Baseline characteristics of these recipients are presented in Table 1. The median age at first transplant was 49 years (IQR 42, 57). Recipients included more females (56%) with a median duration of diabetes at first ITx of 31 years (IQR 23, 41) and insulin requirements of 0.50 units/kg/day (IQR 0.50, 0.70) prior to transplant. Recipients had a median pre-transplant body mass index (BMI) of 25.0 kg/m² (IQR 22.8, 27.7) and eTLV of 1522 cm³ (IQR 1370,1703). Recipients had a median of 2 intraportal infusions (IQR 2, 3), with most infusions taking place within the first year (time to all intra-portal infusions median 323 days (IQR 48, 1560)) (Table 1). 18% (n = 53) received one infusion only, 46% (n = 138) received two infusions within 5.2 months (IQR 2.3, 12) of the first infusion, 22% (n = 66) received three infusions within 42 months (IQR 20, 74), 11% (n = 33) received four infusions within 94 months (IQR 68, 130) and 2.7% (n = 8) received five infusions within 170 months (IQR 150, 190) (Table 1). The majority of infusions used transplant preparations originating from a single donor (97% (n = 673)) with a median total 16871 IE/kg (IQR 12374, 21686) transplanted across all infusions. Transplant preparations had a median purity of 55% (IQR 45, 70), 5% (IQR 3, 13) trapped (in exocrine tissue) islets and a median PCV of 3 mL (IQR 2.5, 4) (Table 2).

### Acute increases in portal venous pressure after intraportal islet infusion are transient

Table 3 summarises number of individual procedures analysed per each infusion number, the median portal pressures pre- and post-infusion, as well as changes in portal pressure per infusion. Portal pressure increased acutely during each transplant procedure (p < 0.001) (Fig 2A) by an overall median 2.0 mmHg (IQR 1.0, 4.0). This acute rise was consistent across subsequent infusions (p > 0.1), with the change in portal pressure at each subsequent infusion not different to the

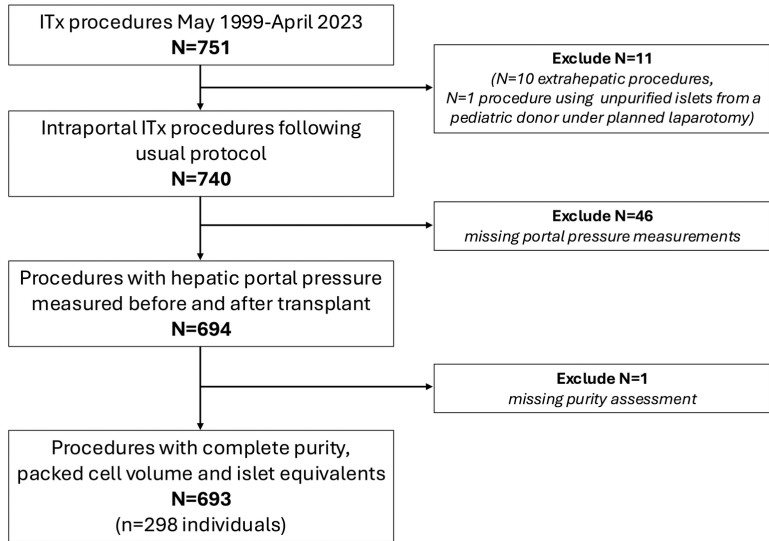

**Fig 1. Inclusion flow chart.**

change after the first infusion, demonstrating no cumulative effect (Fig 2B, Table 3). These acute rises were not sustained, with sequential pre-transplant portal pressures returning to similar levels compared to first transplant (p > 0.1) (Fig 2A).

### Acute increases in portal venous pressure after intraportal islet infusion are mostly explained by PCV, which may be moderated by purity

Initial models assessing individual predictors indicated that PCV exhibited the strongest relationship with changes in portal pressure (Coefficient: 1.0, 95% CI: 0.83, 1.2, p < 0.0001). eTLV (Coefficient: −0.003, 95% CI: −0.0042, −0.0018, p < 0.0001) and purity (coefficient: −0.027, 95% CI: −0.043,-0.011, p = 0.0021) also demonstrated modest yet significant associations with changes in portal pressure. Conversely, IE (per kg) was not a statistically significant predictor and not included in the final modelling (Fig 3). Among the significant predictors- PCV, purity and eTLV - an interaction term between PCV and purity was indicated due to a significant Pearson correlation in initial correlation matrices (R = −0.36, p < 0.0001), indicating that the relationship between PCV and portal pressure may be moderated by purity.

Likelihood ratio tests revealed that a combined model including PCV, purity, eTLV, and the interaction between PCV and purity provided the best fit for predicting changes in portal pressure (ANOVA p < 0.0001), with this model exhibiting the highest performance score with the best relative fit according to the Bayesian/Akaike information criterion (weighted BIC/AIC) (Table 4).

In this final model, higher PCV was independently associated with an increase in portal pressure (coefficient = 1.0 [95% CI 0.81, 1.21]; p < 0.0001), while larger eTLV was associated with a decrease in portal pressure (coefficient = −0.0030 [95% CI −0.0038, −0.0020]; p < 0.0001), when adjusted for other predictors. The interaction term indicated that the effect of PCV on portal pressure decreases as purity increases (coefficient = −0.024 [95% CI −0.033, −0.014]; p < 0.0001). Purity alone was not statistically significant, when adjusted for other predictors (coefficient = 0.0046 [95% CI −0.011, 0.021]; p = 0.580) (Table 5).

This final combined model accounted for 28% of the variance in portal pressure changes between individuals (conditional $R^2$ = 0.277), with 22% explained by the fixed effects alone (PCV, purity, eTLV, and their interaction; marginal $R^2$ = 0.218) (Table 4). The similarity between the model's residual standard deviation (2.90) and RMSE (2.80) (Table 4)

**Table 1. Demographics of islet transplant (ITx) recipients.**

| Characteristic | N = 298[a] |
|---|---|
| Age at first transplant (years) | 49 (42, 57) |
| Sex | |
| *Male* | 131 (44%) |
| *Female* | 167 (56%) |
| Diabetes Duration (years) | 31 (23, 41) |
| *Unknown* | 34 |
| Insulin dose (U/Kg/day) | 0.50 (0.50, 0.70) |
| *Unknown* | 35 |
| Weight (kg) | 72 (64, 81) |
| Height (cm) | 168 (162, 176) |
| *Unknown* | 34 |
| BMI (kg/m$^2$) | 25.0 (22.8, 27.7) |
| *Unknown* | 34 |
| Body Surface Area | 1.83 (1.71, 1.97) |
| *Unknown* | 34 |
| Estimated Total Liver Volume (cm$^3$) | 1,522 (1,370, 1,703) |
| *Unknown* | 34 |
| Number of intra-portal infusions (overall) | 2 (2, 3) |
| Time to all intra-portal infusions (days) | 323 (48, 1,560) |
| Proportion receiving: | |
| *One infusion* | 53 (18%) |
| *Two infusions (Time to infusion in months)* | 138 (46%) (5.2 months (IQR 2.3, 12)) |
| *Three infusions (Time to infusion in months)* | 66 (22%) (42 months (IQR 20, 74)) |
| *Four infusions (Time to infusion in months)* | 33 (11%) (94 months (IQR 68, 130)) |
| *Five infusions (Time to infusion in months)* | 8 (2.7%) (170 months (IQR150, 190)) |

[a] Data are presented as median (IQR) or n (%).

suggests that while the model captures key predictors of portal pressure changes, a portion of variability (residual standard deviation; 2.90) remains unexplained, indicating the potential influence of additional factors.

Exploratory visualisations of the interaction between PCV and purity, using the final combined model and considering the eTLV of an average islet recipient (mean eTLV 1524 cm$^3$), demonstrated that most procedures are likely to result in acute increases in portal pressure between 2–4.5 mmHg (Fig 4).

**The acute portal pressure rise per procedure has remained relatively stable over 24 years of practice in our Centre**

Overall the acute portal pressure rise per procedure has remained relatively stable overtime (Fig 5). However, before 2005 there were a higher proportion of procedures resulting in acute portal pressure rises exceeding 4.5 mmHg (38/119 (31.9%) vs. 77/574 (13.4%); Chi-squared 23.1; p < 0.0001), and indeed more cases using PCV > 5 mL (Fig 5) (36/119 (%) vs. 9/574 (%) Chi-squared 128.8; p < 0.0001).

## Discussion

Our findings provide a comprehensive summary of the changes in portal venous pressure following up to five sequential islet infusions, detailing the portal pressure changes across 693 procedures in 298 recipients within the University of

**Table 2. Islet preparation characteristics for all infusion procedures.**

| Characteristic | N = 693[a] |
|---|---|
| Infusion Type | |
| *Combined islet preparation* | 20 (2.9%) |
| *Single donor islet preparation* | 673 (97%) |
| Islet Equivalents per infusion | 430,798 (366,023, 519,116) |
| Islet Equivalents per Kg per infusion | 6,065 (5,279, 7,085) |
| Total Islet Equivalents received over all infusions | 1,184,345 (853,272, 1,580,289) |
| Total Islet Equivalents per kg received over all infusions | 16,871 (12,374, 21,686) |
| Islet Particle Number per infusion | 393,165 (322,199, 506,585) |
| *Unknown* | 1 |
| Islet Particle Index per infusion | 1.12 (0.94, 1.32) |
| *Unknown* | 1 |
| Purity per infusion (%) | 55 (45, 70) |
| Trapped Islets per infusion (%) | 5 (3, 13) |
| Packed Cell Volume per infusion (mL) | 3.00 (2.50, 4.00) |
| Viability per infusion (%) | 87 (82, 92) |
| *Assay not performed* | 44 |
| Pre-infusion portal pressure (mmHg) | 11.0 (8.0, 13.0) |
| Post-infusion portal pressure (mmHg) | 13.0 (10.0, 16.0) |
| Change in portal pressure (mmHg) | 2.0 (1.0, 4.0) |

[a] Data are presented as median (IQR) or n (%).

**Table 3. Changes in portal pressure per infusion procedure.**

| | Infusion number[a] | | | | |
|---|---|---|---|---|---|
| | 1 | 2 | 3 | 4 | 5 |
| N | 298 | 239 | 107 | 41 | 8 |
| Time to infusion (months) | – | 5.2 (2.3,12) | 42 (20,74) | 94 (68,130) | 170 (150,190) |
| Portal pressure pre-infusion (mmHg) | 11 (8,14) | 11 (8,13) | 11 (8,13) | 12 (9,14) | 12 (10,14) |
| Portal pressure post-infusion (mmHg) | 13 (10,17) | 13 (10,16) | 13 (10,16) | 15 (12,18) | 13 (12,14) |
| **Acute change in portal pressure post-infusion (mmHg)** | **2 (1,3)** | **2 (1,4)** | **2 (0,3)** | **2 (1,4)** | **1 (0,2)** |

[a] Data are presented as median (IQR) or n (%).

Alberta CITP. We observe that acute increases in portal pressure post-infusion are not sustained and rise consistently by an overall median 2 mmHg acutely during each infusion. Importantly we have demonstrated that multiple infusions are not associated with any cumulative effects on portal pressure. Our modelling indicates that 28% of the variability in portal pressure changes between individuals can be attributed to PCV, purity and eTLV, with higher PCV associated with larger increases in portal pressure, while larger eTLV tends to mitigate this increase. Notably, a significant interaction term between PCV and purity indicated that the impact of PCV on portal pressure may be less pronounced at higher islet purities.

Early work from our group, prior to the introduction of intravenous heparin infusion and limited packed cell volume below 5 mL in 2005, highlighted significant increases in the acutely elevated portal pressures in patients requiring more than one infusion [14]. Later studies by Kawahara et al. demonstrated that the implementation of these strategic changes

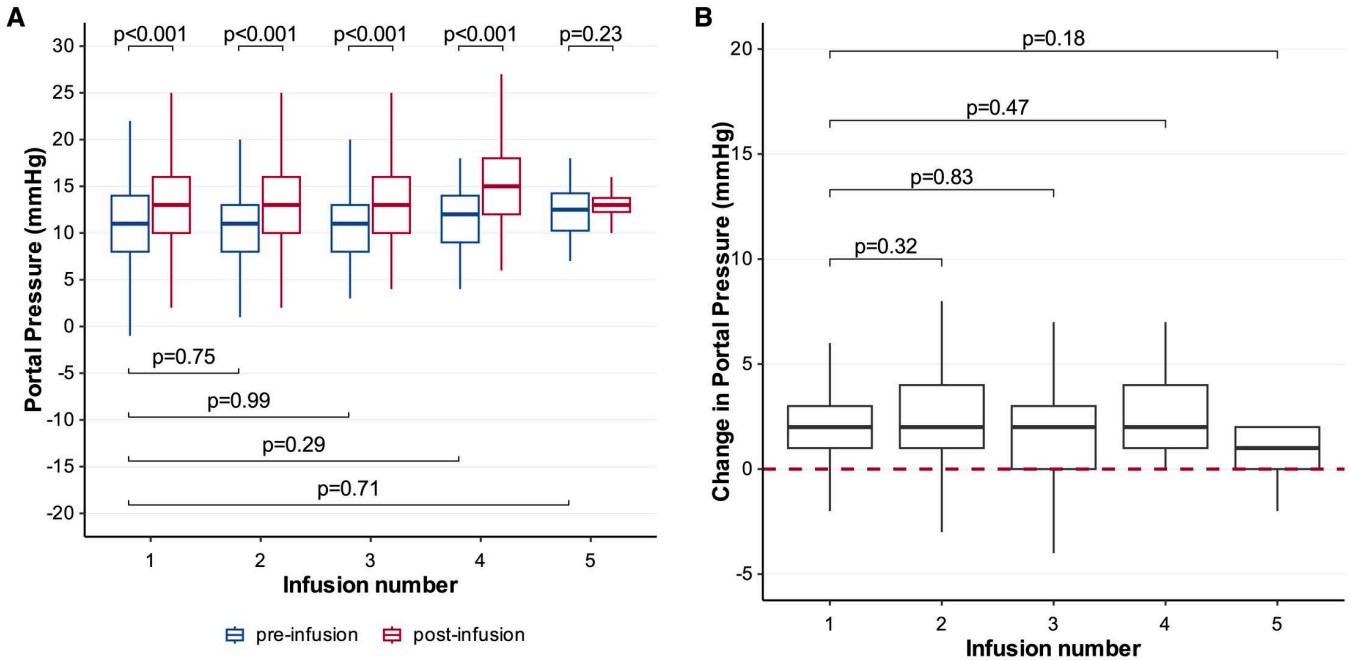

**Fig 2. Box plot of portal pressure before and after each infusion (A) and box plot of overall portal pressure change at each infusion with red dash line indicating zero or no change (B).**

to the procedure were successful in markedly reducing the incidence of portal vein thrombosis [7]. However, this study did note that larger acute increases in portal pressure during transplant procedures was associated with an increased risk of portal vein thrombosis, highlighting an increase above the threshold 4.5 mmHg was predictive of portal vein thrombosis after transplant [7]. Reassuringly, we confirm that acute increases in portal pressure post-infusion are transient, and importantly do not increase with supplementary transplants, with a median 2 mmHg change in portal pressure during each infusion, which was stable for third, fourth or fifth infusions.

Both previous studies have emphasized the role of PCV in determining outcomes after ITx, with higher volumes linked to greater acute elevations of portal pressures [7,14]. Additionally larger livers have been suggested to be more likely to be able to accommodate infusions without substantially elevating portal pressures [7]. We confirm that increases in portal pressures are strongly associated with higher packed cell volumes, with approximately 1 mmHg increase in portal pressure for every 1 mL increase in PCV, when adjusted for the influence of other predictors. Additionally, we demonstrate that the relationship between PCV and portal pressure is influenced by the purity of the islet preparation, identifying a significant interaction term of PCV with purity, with indications that cell the impact of PCV on portal pressure may be less pronounced at higher islet purities. In addition, we also demonstrate a significant association with eTLV, however we note that the effect is small when adjusting for other predictors, highlighting that transplantation in smaller livers may be less of a concern if PCV and purity are appropriately moderated.

Exploratory visualisations of the interaction between PCV and purity, using the final combined model and considering the eTLV of an average islet recipient (mean eTLV 1524 cm³), demonstrated that most procedures are likely to result in acute increases in portal pressure between 2–4.5 mmHg. Notable from this exploratory visualization was that procedures using preparations of low purity could result in increases in portal pressure above 4.5 mmHg, a level which has previously been associated with increased risk for portal vein thrombosis [7], even at the recommended PCV under 5 mL. Highly pure islet preparations are unlikely to cause acute increases in portal pressure exceeding 4.5 mmHg, and at low volume

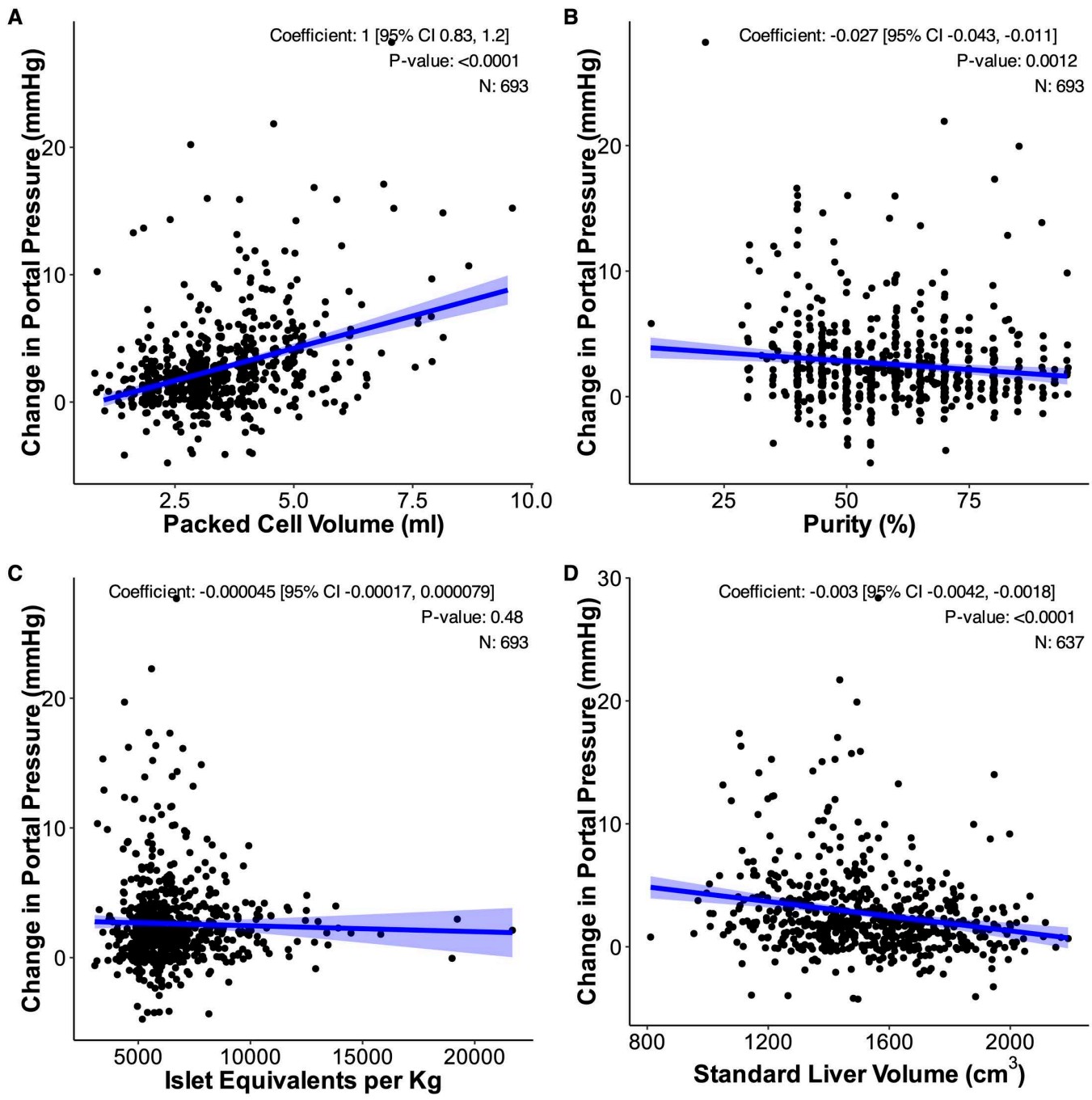

**Fig 3. Scatter graphs of relationship between the acute change in portal pressure and the individual factors: A) Packed cell volume B) Purity of the transplant preparation C) Islet mass per kg and the non-modifiable factor D) Estimated total liver volume. Blue line and shaded area demonstrate the mixed effects model prediction and prediction interval for each factor accounting for individual level differences (random intercept).**

**Table 4. Performance parameters of nested mixed effects models.**

| Model | Conditional R² | Marginal R² | Interclass correlation coefficient | Root mean squared error | Residual standard deviation | Akaike Information Criterion (weighted) | Bayesian Information Criterion (weighted) | Performance Score |
|---|---|---|---|---|---|---|---|---|
| **PCV*purity +eTLV (final model)** | 0.277 | 0.218 | 0.076 | 2.80 | 2.90 | 1.00 | 1.00 | **75.0%** |
| PCV only | 0.272 | 0.143 | 0.150 | 2.73 | 2.91 | $4.33 \times 10^{-10}$ | $3.46 \times 10^{-7}$ | 36.3% |
| Purity+ PCV | 0.272 | 0.143 | 0.150 | 2.73 | 2.91 | $1.60 \times 10^{-10}$ | $1.38 \times 10^{-8}$ | 36.0% |
| Purity+ PCV +eTLV | 0.268 | 0.188 | 0.098 | 2.79 | 2.92 | $3.30 \times 10^{-5}$ | $3.07 \times 10^{-4}$ | 12.4% |

Performance parameters used to compare the 4 nested mixed effects models, in order of performance score.

**Table 5. Summary information of key predictors of portal pressure change included in the final combined mixed effects model.**

| Predictor[a] | Estimate [95% CI] | P-value | Clinical Interpretation |
|---|---|---|---|
| **Packed Cell Volume** | 1.00 [0.81,1.21] | **<0.0001** | Higher PCV is associated with increased portal pressure, with approximately 1 unit increase (mmHg) in portal pressure for every 1 unit (ml) increase in packed cell volume, when adjusted for other predictors. |
| **Purity** | 0.0046 [-0.011,0.021] | 0.580 | When adjusted for other predictors, changes in purity alone do not significantly impact portal pressure. |
| **Packed cell volume:Purity** | −0.024 [-0.033,-0.014] | **<0.0001** | Although the effect size is small, the significant interaction term of PCV with purity suggests that the effect that PCV has on portal pressure decreases as purity increases. |
| **Estimated Total Liver Volume** | −0.0030 [-0.0038,-0.0020] | **<0.0001** | Larger liver volumes are associated with smaller increases, and potentially decreases, in portal pressure while adjusting for other predictors. However, the effect of eTLV on portal pressure is small. |

Coefficients and confidence intervals of key predictors of portal pressure change included in the final combined mixed effects model along with significance of the effect (p-value), emboldened if reaching significance level <0.05, with a suggested clinical interpretation also included. [a]Predictor variables centered to reduce multicollinearity when fitting the final model

are unlikely to cause increases above 4.5 mmHg. It is important to highlight that when isolating islets from deceased donors, there is a trade-off between purity of a preparation and ensuring sufficient islet cells for transplant, additional purification necessarily increases the number of islets lost. Therefore, any predicted effects displayed in this visualization at very high purity should be interpreted with caution due to limitations of the modelling data, since the majority of our preparations have lower purity (median purity of CITP preparations 55% (IQR 45,70). At high purity levels, the perceived dissipation of the green area (<2 mmHg rise) and expansion of the yellow area (2–4.5 mmHg rise) are likely due to minimal predictive capacity at these theoretical levels and we emphasize that this does not imply any recommendation to exceed PCV of 5 mL in clinical settings.

As with all observational studies this report has limitations. The retrospective nature of this analysis captures 24 years of clinical ITx practice within the CITP. Within the early stages of our program (circa 2005), there were important protocol changes implemented specifically to reduce the risks of elevated portal pressure and portal vein thrombosis, in particular therapeutic heparinization and stricter limits surrounding PCV (recommending <5 mL). The implementation of these changes could introduce an era bias effect on changes in portal pressure observed. It is noteworthy that over 24 years of practice in our Centre, the acute portal pressure rise per procedure has remained relatively stable overtime. However, as described, before 2005 there were indeed a higher proportion of procedures resulting in acute portal pressure rises exceeding 4.5 mmHg. This coincided with a higher frequency of cases using PCV > 5 mL occurring also before 2005 – a likely contributing factor to these procedures resulting in acute portal pressure rises exceeding 4.5 mmHg.

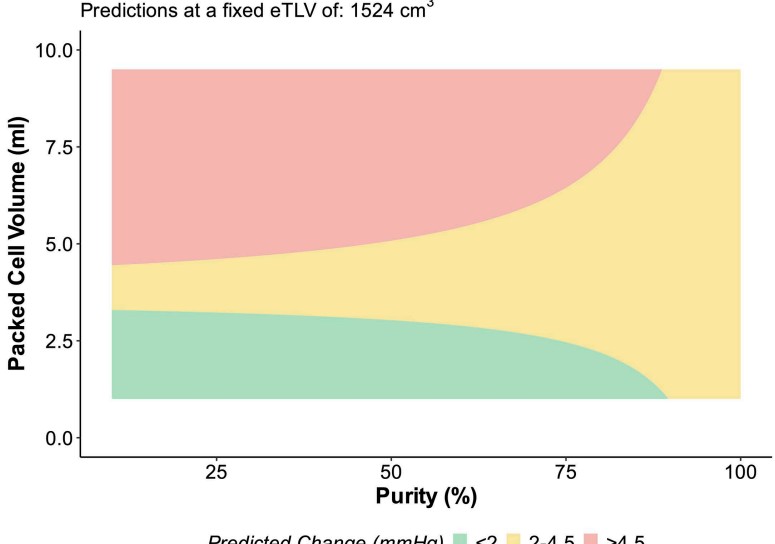

Fig 4.  **Exploratory visualisations of the interaction between PCV and purity using the final model that was derived using CITP data.** The model considers a fixed value of standard lover volume to reflect an average individual, based on the mean eTLV of CITP islet recipients of 1524 $cm^3$. Green area depicts predicted changes in portal pressure: < 2 mmHg, yellow area: 2-4.5 mmHg and red area: > 4.5 mmHg, with 4.5 mmHg being a previously identified threshold for increased risk of portal vein thrombosis [7].

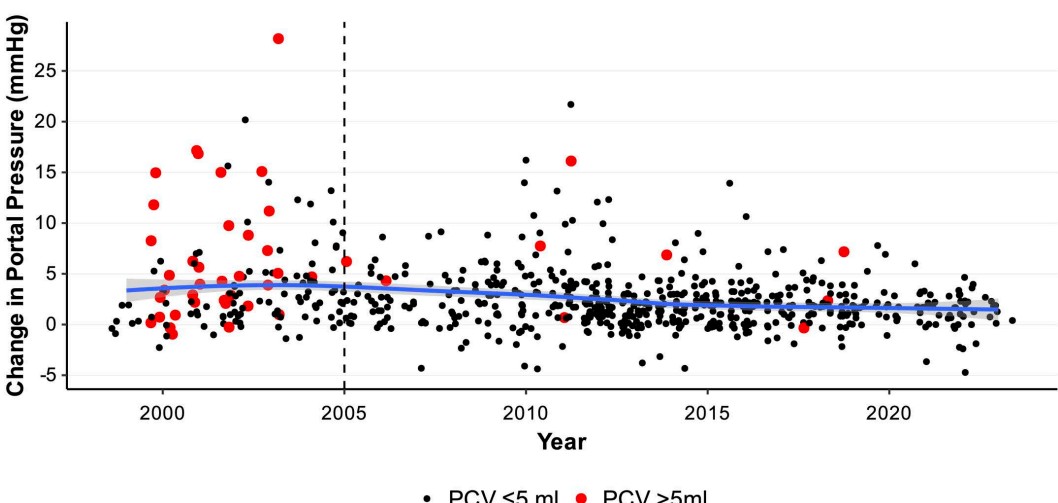

Fig 5.  **Scatter plot of the acute portal pressure rise per case spanning the time of the University of Alberta Clinical Islet Transplant Program (CITP): March 1999-April 2023.** Dashed line indicates the point at which our centre has followed recommendations to maintain a PCV of the transplanted preparation below 5 mL, in addition to maintenance of therapeutic anticoagulation post-ITx with track ablation (2005). Larger red dots highlight cases whose transplant preparation had a PCV exceeding 5 ml.

Furthermore, while we have controlled for several key variables, we identify that these account for only 28% of the variance in portal pressure changes observed between individuals in our cohort. This suggests that there are unaccounted factors that influence portal pressure changes. Speculatively one key, but not easily captured, factor could be the exact portal pressure measurement technique, in particular positioning of the catheter in the portal vein. Furthermore, the

accuracy of the model depends on the precision and consistency of portal pressure measurement, which may have varied across time in our program. Additionally, factors such as individual patient's physiological/ immunological response to the transplant or underlying liver health could introduce unaccounted variability in the outcomes. A limitation is that we do not have detailed information about underlying hepatic pathology at time of every transplant. Early in our program recipients underwent hepatic ultrasound, prior to transplant, immediately post-transplant (days 0 or 1 and day 7), and hepatic ultrasound and MRI approximately one year post-transplant.. Areas of steatosis have been noted in some recipients 1 year post-transplant [26], particularly in individuals with intermediate function, who may need an updated transplant, however no other pathological findings have been identified, such that post-transplant MRI is not routine. Importantly, however, these radiological findings were not detectable on early post-transplant ultrasound examinations conducted within the first week [26]. This suggests that such hepatic changes develop over time and thus would not directly influence the immediate portal pressure measurements obtained immediately before and after transplantation. Transaminases are monitored routinely every 3 months, and have not shown any secular trends (except transient elevations immediately post Tx) [27]. The few liver biopsies performed post-transplant have not shown evidence of fibrosis.

While the predictive models employed in this study provide valuable insights into how PCV, purity and eTLV may influence changes in portal pressure, it is important to note that these models capture the relationships from the data within our single center only. Different centers may have variations in procedure techniques, patient demographics, and post-transplant management practices, which could influence the generalisability to other settings or populations. This is particularly important to highlight when interpreting our exploratory visualisation of the interaction of PCV and purity. While speculative, this could give helpful insight into the optimisation of these parameters. However, this visualisation is for demonstrative purposes and mostly reflects the practice of our centre, since the modelling relationships were derived using CITP data. Furthermore, limitations of the modelling data consequently do not give us a complete picture of the effects of very high levels of purity, since the majority of our preparations have lower purity. Additionally, our data are insufficient to provide definitive statements about the safety of more than five islet infusions but suggest the risk of elevated portal pressure is low. It is also important to highlight that the risk of elevated portal pressure may be higher if the time interval between infusions was shortened substantially, based on an individual's capacity for hepatic regeneration and remodelling in response to embolic portal infusion.

Notwithstanding these limitations, these data provide clinical ITx programs greater insights into factors influencing portal pressure changes during the ITx procedure to optimize safety and minimize risks of portal vein thrombosis. Practically, our data suggest that optimizing islet preparation quality should be prioritized. Patient selection for additional transplants can also be informed by these data, highlighting the importance of careful consideration of eTLV. eTLV has not been widely considered as a risk factor to date, perhaps since ITx is restricted to adults. This may become more important if intraportal ITx was expanded to pediatrics.

Our data may have greater relevance for future stem cell derived islet therapies, which have the potential to treat a much larger proportion of people with diabetes than islets from deceased pancreas donors, particularly if the requirement for lifelong immunosuppression can be removed. Our data may be helpful in addressing potential anxieties regarding repeated intrahepatic delivery necessary to sustain long-term insulin independence. Waning graft function, as a consequence of recurrent autoimmunity, toxicity of immunosuppression, or metabolic burnout, may still necessitate repeated supplementary infusions of stem cell-derived islets. However, stem cell derived islets have no contaminating exocrine components that substantially contribute to the PCV in deceased donor islet preparations. It is therefore anticipated that, from a portal pressure perspective, stem cell derived islet cell therapies will carry minimal acute risk of portal pressure rises which may lead to partial or complete portal thrombosis. These data may help inform discussions with regulators as these new therapies aim for approval as a licensed cell-based therapy for type 1 diabetes, as well as the design of clinical protocols for intraportal transplantation and individual risk:benefit discussions between clinicians and recipients of potentially life transforming therapies.

## Acknowledgments

We acknowledge a large group directly or indirectly involved in the clinical care of patients undergoing pancreatic islet transplantation at the University of Alberta (Edmonton, AB, Canada), surgical fellows, technicians, and support staff from the Clinical Islet Transplant and Give Life Alberta - Organ Donation North (GLAODN) (previously Alberta's Human Organ Procurement and Exchange program). The contributions of organ donors, islet recipients and their families are deeply appreciated.

## Author contributions

**Conceptualization:** Alice Louise Jane Carr, A. M. James Shapiro, Peter A. Senior.

**Data curation:** Alice Louise Jane Carr, Rahi Shah, Braulio A. Marfil-Garza, Doug O'Gorman, Tatsuya Kin.

**Formal analysis:** Alice Louise Jane Carr.

**Investigation:** Alice Louise Jane Carr, Rahi Shah, Braulio A. Marfil-Garza, Anna Lam, Khaled Dajani, Blaire Anderson, Richard T. J. Owen, Doug O'Gorman, Tatsuya Kin, Peter A. Senior.

**Methodology:** Alice Louise Jane Carr, Peter A. Senior.

**Project administration:** Peter A. Senior.

**Supervision:** A. M. James Shapiro, Peter A. Senior.

**Visualization:** Alice Louise Jane Carr.

**Writing – original draft:** Alice Louise Jane Carr.

**Writing – review & editing:** Alice Louise Jane Carr, Rahi Shah, Braulio A. Marfil-Garza, Anna Lam, Khaled Dajani, Blaire Anderson, Richard T. J. Owen, Doug O'Gorman, Tatsuya Kin, David Bigam, A. M. James Shapiro, Peter A. Senior.

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
