## [Decision Letter · Decision Letter 0]

Dear Dr. Carr,

Thank you for submitting your manuscript to PLOS ONE. After careful consideration, we feel that it has merit but does not fully meet PLOS ONE’s publication criteria as it currently stands. Therefore, we invite you to submit a revised version of the manuscript that addresses the points raised during the review process.

We look forward to receiving your revised manuscript.

Kind regards,

Shafiya Imtiaz Rafiqi, PhD

Academic Editor

PLOS ONE

Journal Requirements:

2. We note that your study involved tissue/organ transplantation. Please provide the following information regarding tissue/organ donors for transplantation cases analyzed in your study.

1. Please provide the source(s) of the transplanted tissue/organs used in the study, including the institution name and a non-identifying description of the donor(s).

2. Please state in your response letter and ethics statement whether the transplant cases for this study involved any vulnerable populations; for example, tissue/organs from prisoners, subjects with reduced mental capacity due to illness or age, or minors.

- If a vulnerable population was used, please describe the population, justify the decision to use tissue/organ donations from this group, and clearly describe what measures were taken in the informed consent procedure to assure protection of the vulnerable group and avoid coercion.

- If a vulnerable population was not used, please state in your ethics statement, “None of the transplant donors was from a vulnerable population and all donors or next of kin provided written informed consent that was freely given.”

3. In the Methods, please provide detailed information about the procedure by which informed consent was obtained from organ/tissue donors or their next of kin. In addition, please provide a blank example of the form used to obtain consent from donors, and an English translation if the original is in a different language.

4. Please indicate whether the donors were previously registered as organ donors. If tissues/organs were obtained from deceased donors or cadavers, please provide details as to the donors’ cause(s) of death.

5. Please provide the participant recruitment dates and the period during which transplant procedures were done (as month and year).

6. Please discuss whether medical costs were covered or other cash payments were provided to the family of the donor. If so, please specify the value of this support (in local currency and equivalent to U.S. dollars).

BAMG is a co-inventor for a patent on TNFRSF25-mediated treatments of immune diseases and disorders (PCT/US2020/053085). PAS has received grants from JDRF, CIHR and Brain Canada and consulting or personal fees from Abbott, Bayer, Dexcom, Eli Lilly, GSK, Insulet, Novo Nordisk, Sanofi, Vertex. He served previously as board chair for Diabetes Canada and is a co-lead for Diabetes Action Canada's innovations in type 1 diabetes goal group. AMJS has received grants or contracts from the Breakthrough T1D (formally Juvenile Diabetes Research Foundation), Canadian Stem Cell Network, Diabetes Research Foundation in Canada, ViaCyte, and the US National Institute of Diabetes and Digestive and Kidney Diseases; serves as a consultant to Vertex Inc, Betalin Inc, and  Aspect Biosystems Inc; and is a co-inventor for a patent on TNFRSF25-mediated treatments of immune diseases and disorders (PCT/US2020/053085) and for a Cellular Transplant Site- Device-less technology (US 14/863541, CA.286512). All other authors declare no competing interests.

5. In the online submission form, you indicated that due to the uniqueness of islet transplantation, the relatively low numbers receiving this treatment per year and the known location of the University of Alberta’s Clinical Islet transplant program, there is a risk that the data underlying the results may contain potentially identifying information. Therefore, the data underlying the results reported in this Article are not publicly available. De-identified individual participant data, as well as a data dictionary, can be made available to researchers who provide a methodologically sound proposal with ethics approval. Proposals should be directed to the corresponding author; to gain access, data requestors will need to sign a data access agreement.

Reviewers' comments:

Reviewer's Responses to Questions

**Comments to the Author**

1. Is the manuscript technically sound, and do the data support the conclusions?

Reviewer #1: Yes

Reviewer #2: Yes

2. Has the statistical analysis been performed appropriately and rigorously?

Reviewer #1: Yes

Reviewer #2: Yes

3. Have the authors made all data underlying the findings in their manuscript fully available?

Reviewer #1: Yes

Reviewer #2: Yes

4. Is the manuscript presented in an intelligible fashion and written in standard English?

Reviewer #1: Yes

Reviewer #2: Yes

Reviewer #1: Overall a well written manuscript with a clear and concise message. The cohort studied is the largest and most significant cohort worldwide and therefore the study is of particular significance. The authors make the point clearly that this is of relevance and translational to the field in view of the stem cell islet transplantations that are now happening. It is therefore a useful addition to the literature.

I have some minor comments:

1. line 156 - a little ambiguous - reword

2. Line 164 - I am not certain what is meant by the term random intercept

3. Table 3 - on my version the portal pressure values are not displayed clearly.

4. In the discussion I would solely discuss the results. Please take out Figure 4 and 5 and place these and describe as necessary in the results section. These can then be discussed in the discussion

Reviewer #2: The authors previously reported that an increase in portal vein (PV) pressure (PVP) during the ITx procedure compared to pre- and post- treatment levels. Critical complications such as PV thrombus had occurred when the increased poral vein pressure exceeded 4.5mmHg. In this paper, an acute PVP elevation of less than 2 mmHg was investigated at a lower level. Small fluctuations of PVP were reversible, making their evaluation valuable as a safety indicator based on suitable PVC range and calculated SLV adjustment. This paper is valuable for indicating the safety margin during the PV infusion in ITx. However, a few points should be cleared out to understand the detail for ITx method in this journal.

Minor

#1 PVC information should be cleared out such as basic medium, cell density, dropping speed, viscosity and whether total volume is fixed or not.

#2 To evaluate SLV, the liver function including imaged based assessments should be made clear.

**Do you want your identity to be public for this peer review?** For information about this choice, including consent withdrawal, please see our Privacy Policy

Reviewer #1: No

Reviewer #2: No

---

## [Author Response · Author response to Decision Letter 1]

6 May 2025

We have uploaded a document labeled Response to Reviewers responding point-by-point to both the editor and reviewers requests and comments, including a blank example of the form used to obtain consent from donors as requested.

---

## [Decision Letter · Decision Letter 1]

Dear Dr. Carr,

Thank you for submitting the revised version of your manuscript to PLOS ONE. After careful consideration, we feel that some minor corrections need to be done to fully meet PLOS ONE’s publication criteria.  Therefore, we invite you to submit a revised version of the manuscript that addresses the points raised during the review process.

We look forward to receiving your revised manuscript.

Kind regards,

Shafiya Imtiaz Rafiqi, PhD

Academic Editor

PLOS ONE

Journal Requirements:

Reviewers' comments:

Reviewer's Responses to Questions

**Comments to the Author**

Reviewer #1: All comments have been addressed

Reviewer #3: All comments have been addressed

2. Is the manuscript technically sound, and do the data support the conclusions?

Reviewer #1: Yes

Reviewer #3: Yes

3. Has the statistical analysis been performed appropriately and rigorously?

Reviewer #1: Yes

Reviewer #3: Yes

4. Have the authors made all data underlying the findings in their manuscript fully available?

Reviewer #1: No

Reviewer #3: Yes

5. Is the manuscript presented in an intelligible fashion and written in standard English?

Reviewer #1: Yes

Reviewer #3: Yes

Reviewer #1: I am satisfied all comments have been addressed and legitimate reasons for not making data publicly available is entirely appropriate.

Reviewer #3: The study provides valuable insights into the determinants of acute portal pressure changes during sequential islet transplantation. However, the discussion would benefit from a clearer articulation of the clinical implications of these findings. For instance, how might this evidence guide real-world decisions regarding PCV optimization, infusion scheduling, or patient selection for additional transplants? Providing a more explicit set of practice recommendations—derived from your data—would significantly enhance the translational impact of the study.

The authors have effectively demonstrated that estimated total liver volume (eTLV) is a modest predictor of portal pressure changes. However, the potential influence of underlying hepatic pathology—such as steatosis, fibrosis, or other structural alterations—on portal pressure response has not been addressed. Since these conditions may modulate intrahepatic vascular compliance, please consider discussing whether liver health status was assessed and, if not, acknowledge this as a limitation. Inclusion of even indirect indicators (e.g., transaminases, Fib-4 score) could strengthen the interpretation.

**Do you want your identity to be public for this peer review?** For information about this choice, including consent withdrawal, please see our Privacy Policy

Reviewer #1: No

Reviewer #3: No

---

## [Author Response · Author response to Decision Letter 2]

7 Jul 2025

We would like to highlight the update to reference 16 to be, considering the recent journal article release during the American Diabetes Association Conference in June 2025:

16. Reichman TW, Markmann JF, Odorico J, Witkowski P, Fung JJ, Wijkstrom M, et al. Stem Cell–Derived, Fully Differentiated Islets for Type 1 Diabetes. New England Journal of Medicine. 0. doi:10.1056/NEJMoa2506549

Reviewer #3: The study provides valuable insights into the determinants of acute portal pressure changes during sequential islet transplantation. However, the discussion would benefit from a clearer articulation of the clinical implications of these findings. For instance, how might this evidence guide real-world decisions regarding PCV optimization, infusion scheduling, or patient selection for additional transplants? Providing a more explicit set of practice recommendations—derived from your data—would significantly enhance the translational impact of the study.

We appreciate the reviewer’s suggestion to clearly articulate clinical implications. Indeed, throughout our discussion, we explicitly address the practical relevance of our findings for clinical decision-making, emphasizing reassurance around the safety of repeated intrahepatic islet infusions and highlighting the importance of optimizing islet preparation quality (specifically purity and PCV) and careful consideration of liver volume (eTLV) during patient selection. We have also clearly emphasized that our results may help alleviate concerns related to repeated intrahepatic deliveries, particularly relevant for future stem-cell therapies for type 1 diabetes.

Historical experiences with unpurified islet preparations were associated with acute rises in portal pressure, and portal vein thrombosis. We aimed to provide reassurance that repeated intrahepatic infusions of deceased donor islets can be performed safely even in people with a range of liver sizes. Anecdotally, in the field of type 1 diabetes and islet transplantation, there has been anxiety about repeating intrahepatic infusions that may be necessary to maintain long-term insulin independence with future stem cell therapies.

We add additional sentences:

Line 494-497 Practically, our data suggest that optimizing islet preparation quality should be prioritized. Patient selection for additional transplants can also be informed by these data, highlighting the importance of careful consideration of eTLV.

Line 503-505 Our data may be helpful in addressing potential anxieties regarding repeated intrahepatic delivery necessary to sustain long-term insulin independence.

We have deliberately refrained from providing explicit clinical recommendations due to the observational, single-center nature of our data, as discussed within our limitations. We acknowledge potential constraints regarding the generalizability of our findings to other centers or patient populations.

Reviewer #3: The authors have effectively demonstrated that estimated total liver volume (eTLV) is a modest predictor of portal pressure changes. However, the potential influence of underlying hepatic pathology—such as steatosis, fibrosis, or other structural alterations—on portal pressure response has not been addressed. Since these conditions may modulate intrahepatic vascular compliance, please consider discussing whether liver health status was assessed and, if not, acknowledge this as a limitation. Inclusion of even indirect indicators (e.g., transaminases, Fib-4 score) could strengthen the interpretation.

Thank you for highlighting this important consideration. We agree that underlying hepatic pathology, such as steatosis, fibrosis, or other structural changes, could potentially influence portal pressure responses. A limitation of our analysis is indeed the lack of comprehensive hepatic pathology assessments at the time of every transplant, . We have hence expanded on our comment surrounding lack of liver health information:

Line 458-472: A limitation is that we do not have detailed information about underlying hepatic pathology at time of every transplant. Early in our program recipients underwent hepatic ultrasound, prior to transplant, immediately post-transplant (days 0 or 1 and day 7), and hepatic ultrasound and MRI approximately one year post-transplant.. Areas of steatosis have been noted in some recipients 1 year post-transplant [26], particularly in individuals with intermediate function, who may need an updated transplant, however no other pathological findings have been identified, such that post-transplant MRI is not routine. Importantly, however, these radiological findings were not detectable on early post-transplant ultrasound examinations conducted within the first week [26]. This suggests that such hepatic changes develop over time and thus would not directly influence the immediate portal pressure measurements obtained immediately before and after transplantation. Transaminases are monitored routinely every 3 months, and have not shown any secular trends (except transient elevations immediately post Tx) [27]. The few liver biopsies performed post-transplant have not shown evidence of fibrosis.

---

## [Editor Report · Decision Letter 2]

Sequential Islet Transplants for Type 1 Diabetes Are Not Associated with Sustained or Cumulative Increases in Hepatic Portal Venous Pressure

PONE-D-25-03511R2

Dear Dr. Carr,

We’re pleased to inform you that your manuscript has been judged scientifically suitable for publication and will be formally accepted for publication once it meets all outstanding technical requirements.

Kind regards,

Shafiya Imtiaz Rafiqi, PhD

Academic Editor

PLOS ONE
---

## [Editor Report · Acceptance letter]

PONE-D-25-03511R2

PLOS ONE

Dear Dr. Carr,

I'm pleased to inform you that your manuscript has been deemed suitable for publication in PLOS ONE. Congratulations! Your manuscript is now being handed over to our production team.

Kind regards,

on behalf of

Dr. Shafiya Imtiaz Rafiqi

Academic Editor

PLOS ONE